# Comparative Study on the Characterization of Myofibrillar Proteins from Tilapia, Golden Pompano and Skipjack Tuna

**DOI:** 10.3390/foods11121705

**Published:** 2022-06-10

**Authors:** Huibo Wang, Zhisheng Pei, Changfeng Xue, Jun Cao, Xuanri Shen, Chuan Li

**Affiliations:** 1Hainan Provincial Engineering Research Centre of Aquatic Resources Efficient Utilization in the South China Sea, School of Food Science and Engineering, Hainan University, Haikou 570228, China; 19085231210037@hainanu.edu.cn (H.W.); peizhis@hntou.edu.cn (Z.P.); 992985@hainanu.edu.cn (J.C.); 990497@hainanu.edu.cn (X.S.); 2School of Food Science and Engineering, Hainan Tropical Ocean University, Sanya 572022, China; xuecf@hntou.edu.cn; 3Collaborative Innovation Center of Provincial and Ministerial Co-Constructin for Marine Food Deep Processing, Dalian Polytechnic University, Dalian 116034, China

**Keywords:** myofibrillar proteins, physicochemical properties, conformational structure, functional properties, N-glycosylation

## Abstract

In this study, the physicochemical properties, functional properties and N-glycoproteome of tilapia myofibrillar protein (TMP), golden pompano myofibrillar protein (GPMP) and skipjack tuna myofibrillar protein (STMP) were assessed. The microstructures and protein compositions of the three MPs were similar. TMP and GPMP had higher solubility, sulfhydryl content and endogenous fluorescence intensity, lower surface hydrophobicity and β-sheet contents than STMP. The results showed that the protein structures of TMP and GPMP were more folded and stable. Due to its low solubility and high surface hydrophobicity, STMP had low emulsifying activity and high foaming activity. By N-glycoproteomics analysis, 23, 85 and 22 glycoproteins that contained 28, 129 and 35 N-glycosylation sites, were identified in TMP, GPMP and STMP, respectively. GPMP had more N-glycoproteins and N-glycosylation sites than STMP, which was possibly the reason for GPMP’s higher solubility and EAI. These results provide useful information for the effective utilization of various fish products.

## 1. Introduction

As one of the essential protein sources, fish account for 18% of the human protein supply [1]. Fish proteins are classified as salt-soluble proteins (myofibrillar protein (MP)), water-soluble proteins (sarcoplasmic reticulum), and insoluble proteins (matrix protein) according to their solubility [2]. Occupying the highest fish protein content, the MP participates in the regulation and contraction of the muscle and plays a vital role in the quality of fish products [3]. It was reported that the conformation of MP affects the functional properties of fish products [4]. MP has been extracted from different sources such as tuna, silver carp and cod [5,6,7]. There are differences in the protein conformation and properties of MP from different sources. However, MP research often focuses on the effects of different processes, and only a few studies have compared the MPs from different sources [8]. Wang et al. investigated the differences in the rheological properties of MP extracted from fish, beef, sheep and pork [9]. The previous studies reported the differences in physicochemical, functional and rheological properties of MP extracted from beef, lamb, chicken, tuna [10]. Most of the studies performed comparisons on MP from fish and mammals, but few studies characterized the physicochemical properties and functional properties of MP extracted from different fish species.

Tilapia (*Oreochromis niloticus*) is one of the main freshwater fish cultured in the world because of its high nutritional value and easy cultivation, and the tilapia production in China was 1.65 million tons in 2020 [11]. As a vital economic marine fish, the golden pompano (*Trachinotus ovatus*) has the advantages of fast growth and wide salt tolerance and has abundant production in the South China Sea [12]. Skipjack tuna (*Katsuwonus pelamis*) is an essential global seawater fish resource, accounting for more than 70% of tuna production [13]. Skipjack tuna is usually canned due to its strong fishy taste and dark muscle. In addition, the production of all three species of fish is high in the tropical area of China.

Post-translational modifications of proteins, especially their glycosylations, are essential to proteins’ structures and physicochemical and functional properties [14]. N-glycoproteomics has been used to identify N-glycosylation sites of chicken egg white, providing essential information for understanding its glycoproteins’ structure, function and biological activity [15]. Yang et al. compared the N-glycoproteins and their N-glycosylation sites in whey components of mammals, including humans and cattle breeds [16]. In addition, N-glycans can confer protein stability through intramolecular protein-carbohydrate interactions. However, the identification and comparison of MPs’ N-glycoproteins from different fish breeds has not been studied.

In this study, the tilapia myofibrillar protein (TMP), golden pompano myofibrillar protein (GPMP) and skipjack tuna myofibrillar protein (STMP), and their physicochemical properties, protein conformation and functional properties were characterized. The identification of N-glycoproteins of three MPs was analyzed using a liquid chromatography separation and mass spectrometric (LC-MS/MS). Then the N-glycosylation sites of three MPs were identified based on the MS/MS data. In addition, the differences in MPs from different fish species were analyzed for the first time via the N-glycoproteome. The results of this study provide useful information for the rational utilization and product development of various fish resources and the expansion of N-glycosylated sites and N-glycoproteins.

## 2. Materials and Methods

### 2.1. Materials

Fresh tilapia fillets (150 ± 20 g) and golden pompano fillets (100 ± 20 g) were provided from Xiangtai Fishery Co., Ltd. (Chengmai, Hainan, China). Fresh skipjack tunas (1500 ± 20 g) were purchased from a local market (Haikou, China).

### 2.2. Extraction of MPs and Preparation of MP Solutions

The MP was extracted as the previous study with minor modifications [5]. Briefly, the red muscle was removed from the fillets and the remaining white muscle was ground using a QSJ-B02R1 commercial grinder (Bear Co., Ltd., Shanghai, China) in ice water. The minced white muscle was rinsed three times in a low phosphate buffer solution (0.05 mol/L NaCl, 3.38 mmol/L NaH_2_PO_4_·2H_2_O, 15.5 mmol/L Na_2_HPO_4_·12H_2_O, pH 7.5) at 4 °C. Subsequently, the precipitates of tilapia and golden pompano were centrifuged at 7000× *g* using a refrigerated centrifuge (X1R, Thermo, Osterode, Germany) in high phosphate buffer solution (0.6 mol/L NaCl, 3.38 mmol/L NaH_2_PO_4_·2H_2_O, 15.5 mmol/L Na_2_HPO_4_·12H_2_O, pH 7.0); the precipitate of skipjack tuna was centrifuged at the same condition in high phosphate buffer solution (1.5 mol/L NaCl, 3.38 mmol/L NaH_2_PO_4_·2H_2_O, 15.5 mmol/L Na_2_HPO_4_·12H_2_O, pH 7.0) and then stored at 4 °C for 24 h. The mixture was centrifuged for 10 min at 16,000× *g*, the supernatant was collected in the cold distilled water (5-fold) and precipitated at 4 °C for 30 min. The precipitate (MP) was collected after two more centrifugations at 16,000× *g* for 15 min. The yield of the extracted MP was 26.90 ± 1.37% and the concentrations of tilapia myofibrillar protein (TMP), golden pompano myofibrillar protein (GPMP) and skipjack tuna myofibrillar protein (STMP) was 11.38 ± 0.69%, 18.78 ± 0.93% and 10.23 ± 0.52%, respectively. The yield of MP was calculated according to the following formula.
(1)$$\mathrm{The}\;\mathrm{yield}\;\mathrm{of}\;\mathrm{MP}\;\left(\%\right)=\frac{m_{1}}{m_{2}}\times 100$$
where *m*_1_ = the wet mass of extracted MP (g) and *m*_2_ = the wet mass of minced white muscle (g).

The protein concentration was determined according to the method described by the previous study [17], using bovine serum albumin as a standard.

To prepare MP solutions, the obtained MP was dispersed in distilled water (0.6 mol/L NaCl, pH 7.0), then stored at 4 °C until further use.

### 2.3. Sodium Dodecyl Sulfate-Polyacrylamide Gel Electrophoresis (SDS-PAGE)

SDS-PAGE was performed following the previous study using a precast SDS-PAGE gel (BeyoGel™ Plus PAGE, Biyuntian Biotechnology, Shanghai, China) consisting of 4.0% polyacrylamide stacking gels and 10.0% polyacrylamide separating gels [18]. Coomassie Brilliant Blue R250 was used to stain the protein, and the electrophoresis pattern was analyzed with Image Lab software (Bio-Rad Laboratories, Inc., Berkeley, CA, USA).

### 2.4. Scanning Electron Microscope (SEM)

The MP solution (5 mg/mL) was pre-frozen at −80 °C and then lyophilized for 12 h. The dehydrated samples were coated with gold particles and analyzed using the SEM (JSM-7610F Plus, Hitachi, Tokyo, Japan) at an accelerating voltage of 10 kV.

### 2.5. Protein Solubility

The measurement of protein solubility was according to the previous method [19]. The 10 mL of MP solution (5 mg/mL) was centrifuged for 10 min at 10,000× *g* at 4 °C. The ratio of protein concentration in the supernatant to total protein content in the sample determined by the Biuret method represented protein solubility.

### 2.6. Surface Hydrophobicity

Determining surface hydrophobicity referred to Zhou’s method with minor modifications [20]. The 10 mL of MP solution (5 mg/mL) was mixed with 600 μL of bromophenol blue (BPB) solution (1 mg/mL) and vortexed for 5 min. The control sample was prepared using the buffer solution (0.6 mol/L NaCl, 3.38 mmol/L NaH_2_PO_4_·2H_2_O, 15.5 mmol/L Na_2_HPO_4_·12H_2_O, pH 7.5). The MP and control samples were centrifuged for 15 min at 16,000× *g*, and the supernatant was determined for absorbance at 595 nm. The surface hydrophobicity was calculated using the following formula:(2)$$\mathrm{Surface}\;\mathrm{hydrophobicity}\;\left(\mu g\right)=600\;\mu L\times \frac{\left(OD_{1}-OD_{2}\right)}{OD_{1}}$$
where *OD*_1_ and *OD*_2_ are the respective absorbance value of blank control and test sample.

### 2.7. Total and Reactive Sulfhydryl (SH) Group Content

The total SH and reactive SH content of MP were determined according to a previous study with minor modifications [21].

The total SH content: the 1.5 mL of MP solution (5 mg/mL) was suspended in 10 mL Tris-Glycine buffer (0.086 mol/L Tris, 0.09 mol/L Glycine, 4 mmol/L EDTA, 8 mol/L urea, pH 8.0); the reactive SH content: the 1.5 mL of MP solution (5 mg/mL) was mixed with 10 mL Tris-Glycine buffer (0.086 mol/L Tris, 0.09 mol/L Glycine, 4 mmol/L EDTA, pH 8.0). Subsequently, to each treated sample was added 50 μL Ellman reagent (4 mg/mL DTNB), followed by centrifugation at 16,000× *g* for 10 min. The absorbance at 412 nm of the supernatant was recorded using samples without DTNB as the control group. The total and reactive SH contents were calculated as follows:(3)$$\mathrm{SH}\;\mathrm{content}\;\left(\mu \mathrm{mol}/g\right)=73.53\times \frac{A}{\rho }$$
where *A* = absorbance at 412 nm, 73.53 = 106/1.36104, 1.36104 is the molar extinction coefficient (cm/mol) and *ρ* = protein mass concentration of the sample (mg/mL).

### 2.8. Secondary Structure

The secondary structure of MP was measured by using a Fourier transform infrared spectroscopy (FTIR) [22]. After freeze-drying, the sample was mixed with KBr and pressed into a thin pellet. The pellet was scanned in the range of 400–4000 cm^−1^, at a resolution of 4 cm^−1^ for 32 cans. The secondary structure of MP was calculated by using PeakFit v4.12 software to analyze the amide I band components.

### 2.9. Tertiary Structure

The tertiary structure of MP was determined using a fluorescence spectrophotometer. The MP solution (6 mg/mL) was centrifuged at 10,000× *g* for 10 min. The intrinsic fluorescence intensity of the supernatant was analyzed under an excitation wavelength of 280 nm and an emission spectrum of 300–400 nm at a rate of 120 nm/min.

### 2.10. Functional Properties

The emulsifying activity index (EAI) and emulsifying stability index (ESI) were determined according to a previous study with some modifications [23]. A total of 15 mL MP solution (10 mg/mL) and 5 mL corn oil were homogenized at 8000 rpm for 2 min. A 50 μL of the emulsion was taken from the bottom at 0 and 30 min, and mixed with 0.1% SDS solution. The absorbance was recorded at 500 nm. The EAI and ESI were calculated as the following equations:(4)$$\mathrm{EAI}\;\left(m^{2}/g\right)=\frac{2\times 2.303\times A_{0}\times N}{c\times \emptyset \times \left(1-\theta \right)\times 10,000}$$
(5)$$\mathrm{ESI}\;\left(\min\right)=\left(A_{0}\times \Delta t\right)/(A_{0}-A_{30})$$
where *A*_0_ and *A*_30_ are the absorbances at 0 min and 30 min, *N* is the dilution factor, *φ* is the optical path, *θ* is the fraction of oil, *c* is protein concentration, and Δ*t* = 30 min.

The foaming ability (FA) and foaming stability (FS) were measured according to Tao’s method [24]. MP was diluted to 70 mL of 1 mg/mL MP solution in a sample bottle and the height of the solution was recorded as *V*_0_. The solution was homogenized at 10,000 rpm for 2 min, recording the height of the foam as *V*_1_. After standing for 30 min, the height of the foam was recorded as *V*_30_. The FA and FS were calculated as the following equations:(6)$$\mathrm{FA}\left(\%\right)=\frac{V_{1}}{V_{0}}\times 100$$
(7)$$\mathrm{FS}\left(\%\right)=\frac{V_{30}}{V_{1}}\times 100$$

### 2.11. Digestion, Enrichment and Deglycosylation of MP Glycopeptides

Protein digestion was performed using the filter-aided sample preparation (FASP) method as in previous work [25]. Briefly, MP was diluted with UA buffer (8 mol/L urea, 0.1 mol/L Tris-HCl, pH = 8.0) to a volume of 200 μL in an ultrafiltration tube. Then the protein was alkylated with 0.05 mol/L iodoacetamide and incubated in the dark for 20 min. After centrifugation at 14,000× *g* for 15 min, the sample was rinsed three times with 40 mmol/L NH_4_HCO_3_. As for enzymatic digestion, MP was digested with 4 μg trypsin in 40 μL 40 mmol/L NH_4_HCO_3_ at 37 °C overnight. After centrifugation, the tryptic peptides were transferred to ultrafiltration tubes and incubated for 1 h, with a lectin mixture containing concanavalin A, wheat germ agglutinin and Ricinus communis agglutinin in 2× binding buffer [16]. The lectin-bound peptides were rinsed 4 times with 200 μL binding buffer and twice with 50 μL 40 mmol/L NH_4_HCO_3_ in 18O water. For deglycosylation, a total of 2 μL PNGase F (1 U/μL 18O water) in 40 μL 40 mmol/L NH_4_HCO_3_ was added to peptides and then incubated for 3 h at 37 °C. The deglycosylated peptides were rinsed twice with 50 μL 40 mmol/L NH_4_HCO_3_. Then the peptides were collected and lyophilized for further analysis.

### 2.12. Liquid Chromatography Separation and Mass Spectrometric (LC-MS/MS) Analysis

LC-MS/MS was performed using a Q Exactive Plus mass spectrometer coupled with Easy 1200 nLC (ThermoFisher, Waltham, MA, USA). The peptides were diluted with 0.1% formic acid and loaded into a trap column. The reverse-phase high-performance liquid chromatography (RP-HPLC) mobile phase A was 0.1% formic acid in the water, and B was 0.1% formic acid in 95% acetonitrile at a flow rate of 300 nL/min. After separation, the peptides were analyzed. MS data were acquired using a data-dependent top20 method, dynamically choosing the most abundant precursor ions from the survey scan (350–1800 *m*/*z*) for higher-energy collisional dissociation (HCD) fragmentation. The full MS scans were acquired at a resolution of 70,000 at *m*/*z* 200 for MS/MS scan. The maximum injection time was set to 50 ms for MS and 50 ms for MS/MS and the dynamic exclusion duration was 60 s.

### 2.13. Sequence Database Searching and Data Analysis

The analysis of MS data was performed using MaxQuant software (version 1.6.1.0), searching against the databases (UniProt-Oreochromis-136238-20210430.fasta, UniProt-Carangidae (jacks) [8157]-68877-20210430.fasta, and UniProt-Gobiiformes (gobies and sleepers)-46124-20210430.fasta) from the UniProt database. The database search results were filtered and exported with <1% false discovery rate (FDR) at the site level. Other parameter settings were according to the previous study [26].

### 2.14. Statistical Analysis

All experiments were repeated three times. SPSS (version 13.0, SPSS Inc., Chicago, IL, USA) was used for statistical analysis by one-way analysis of variance (ANOVA) at *p* < 0.05.

## 3. Results and Discussion

### 3.1. Protein Components of MPs

It is generally accepted that protein component plays a vital role in the quality of fish products [27]. As shown in Figure 1A, the protein composition of the three MPs was similar, and all consisted of myosin heavy chain (MHC; 220 kDa), actin (44 kDa), tropomyosin (37 kDa) and myosin light chain (MLC; 20 kDa). Compared to TMP, the bands of MHC in GPMP and STMP were more intensive. Meanwhile, myosin aggregations were found at the top of gels, especially STMP indicating a large number of myosin aggregates held by disulfide bonds. The previous study supported the results [20]. Additionally, there was a higher content of tropomyosin in STMP than in TMP and GPMP; it was possible that living in a high concentration of salt made skipjack tuna and golden pompano possess the higher contents of salt soluble protein than tilapia.

### 3.2. Microstructures of MPs

It was observed that the three MPs all exhibited the fibrous structure with some granular protrusions on the surface at the magnification of ×4000 (Figure 1B). At the higher magnification of ×25,000, plenty of irregular protein particles were assembled in TMP and GPMP. However, the aggregation of more compact protein particles resulted in a coarser fibrous structure in STMP.

### 3.3. Solubility of MPs

Solubility plays a significant role in the functional property of protein and reflects the denaturation and aggregation of the protein. In addition, the intermolecular forces such as hydrophobic force and disulfide bond were generally considered to be related to the solubility [19]. As shown in Figure 2A, there was a significant difference in the solubility of the three MPs with the NaCl concentration of 0.6 M at pH 7.0 (*p* < 0.05). The solubility of TMP (69.75%) was higher than GPMP (45.14%) and STMP (12.72%). Additionally, the solubility of TMP was close to the cod proteins in the previous study [28]. The solubility of STMP extracted from red muscle was significantly lower than TMP and GPMP extracted from white muscle. Similarly, the previous study has reported that mackerel MP extracted from red muscle has a lower solubility than MP extracted from white muscle [29]. The lower solubility of STMP was probably due to its high surface hydrophobicity, which would cause protein aggregation and reduce the solubility.

### 3.4. Surface Hydrophobicity and Tertiary Structure of MPs

Surface hydrophobicity reflects the emulsifying properties and is related to the solubility of the protein. The more BPB bound to the myosin molecules, the more hydrophobic sites on the protein structure [30]. The surface hydrophobicity of STMP was significantly higher than TMP and GPMP (*p* < 0.05, Figure 2B). It is known that the native structure of proteins tends to bury their hydrophobic core and distribute hydrophilic groups on the surface. The higher bound BPB value demonstrated the unfolding of the MP structure, thus exposing more non-polar amino acid residues [31]. A previous study reported similar results that the surface hydrophobicity of the tuna sample was the highest among all samples [10]. Strong hydrophobic interaction was generally considered the reason for protein aggregation [32]. STMP was more prone to aggregate due to its high surface hydrophobicity, which explained the coarse fibrous structure of STMP in SEM (Figure 1B). As for myosin, the surface hydrophobicity was inversely proportional to its solubility [33,34]. According to Figure 2A,B, the strong hydrophobic force made STMP particles aggregate more compact, thus greatly lowering their solubility.

The intrinsic fluorescence intensity is related to hydrophobic residues such as tryptophan and tyrosine, and the intrinsic fluorescence spectrum reflects the changes in the tertiary structure of the protein [35]. The unfolding of the tertiary structure of MP leads to the exposure and oxidation of tryptophan residues, resulting in changes in the microenvironment and decreased fluorescence intensity [36]. As shown in Figure 2D, compared with GPMP and TMP, the fluorescence intensity of STMP was the lowest (*p* < 0.05), which indicated that the tertiary structure of MP was unfolded. The intrinsic fluorescence intensity is related to hydrophobic residues such as tryptophan and tyrosine, and the intrinsic fluorescence spectrum reflects the changes in the tertiary structure of the protein. The maximum emission wavelength (λ_max_) of the three MPs exceeded 330 nm, indicating that part of tryptophan residues was in the polar environment [24]. Additionally, more exposures to hydrophobic amino acid residues due to protein unfolding also led to higher surface hydrophobicity, resulting in protein aggregation [36].

### 3.5. Total and Reactive SH Group Content of MPs

Protein folding and unfolding and disulfide bond formation can be reflected by SH group content, which is closely related to the tertiary and quaternary structure of the proteins [28]. Figure 2C shows the total and reactive SH contents of the three MPs. Compared to TMP and GPMP, it was found that the total and reactive SH contents of STMP were the lowest (2.86 ± 0.22 μmol/g). It was found that the SH group contents of all three MPs were lower than the results from grass carp in the previous study, which was probably due to the differences in species [37]. There was a slight difference in total SH content, and no significant difference in active SH content between TMP and GPMP (*p* > 0.05). This phenomenon might be explained because STMP was fully unfolded with the NaCl concentration of 0.6 M at pH 7.0, exposing more reactive SH groups. The unfolding of STMP exposed more SH groups and more internal hydrophobic groups resulting in its higher surface hydrophobicity. However, the exposed reactive SH groups would cross-link to form disulfide bonds to reduce the SH contents. Bu, et al. [38] also attributed the reduction in SH content in southern bluefin tuna during refrigerated storage to disulfide bond formation. Myosin aggregates formed by disulfide bonds as the leading force were hard to solubilize [39]. Therefore, more myosin aggregated held by disulfide bonds in STMP also led to the protein aggregation at the top of the electrophoretic pattern and its lowest solubility.

### 3.6. Secondary Structure of MPs

The FTIR spectra and the secondary structure of MP were showed in Figure 3. Results in Figure 3A showed that there were similar peaks, including amide A (3413 cm^−1^), amide B (2927 cm^−1^), amide I (1640 cm^−1^), amide II (1537 cm^−1^) and amide III (1240 cm^−1^) in three MPs with slight differences. It is known that the amide I region (1600–1700 cm^−1^) is a vital band for analyzing protein secondary structure confirmation [40]. After deconvolution of the amide I, the four kinds of conformation of the protein secondary structure were obtained as α-helix (1650–1660 cm^−1^), β-sheet (1600–1640 cm^−1^), β-turn (1660–1695 cm^−1^) and random coil (1640–1650 cm^−1^), respectively [21].

As shown in Figure 3B, the components of the protein secondary structure showed differences due to different fish species. The major secondary structures in the three MPs were β-sheet and α-helix. However, the ratio of β-sheet in STMP (57.84%) was higher than TMP and GPMP (*p* < 0.05). Meanwhile, the ratio of α-helix in TMP (16.01%) was higher than GPMP and STMP (*p* < 0.05). The ratios of β-sheet and α-helix reflected the changes in protein secondary structure. More hydrophobic sites would become expose because of the lack of α-helix in STMP, which was consistent with the results that STMP had a higher bound BPB value. The low percentage of β-sheet in TMP and GPMP (35.49% and 47.04%, respectively) indicated the unfolding extent of MP structure was low and protein was in a stable state [19].

### 3.7. Functional Properties of MPs

EAI and ESI usually evaluate the emulsifying properties of the food protein. There was no significant difference in ESI and a slight difference in EAI between TMP and GPMP (Figure 4A). Compared to TMP and GPMP, the EAI (122.06 m^2^/g) and ESI (30.96 min) of STMP were the lowest among the three MPs. Generally, the surface hydrophobicity of proteins is considered the critical factor affecting their emulsifying properties. The exposure of internal hydrophobic groups and sulfhydryl groups due to the unfolding of the protein structure would strengthen the surface hydrophobicity and the formation of disulfide bonds. The enhanced interaction might improve steric stability against the flocculation and aggregation of oil droplets [28]. However, even though the value of surface hydrophobicity of STMP was the highest and reactive SH content of STMP was the lowest, interestingly it was found that the emulsifying properties of STMP were not improved with more hydrophobic sites. The same phenomenon was found in Yan’s study that the phosphorylated walnut protein exhibited the lowest surface hydrophobicity but the highest emulsifying activity [41]. It was reported that excessive surface hydrophobicity affected the surface stability of oil droplets, making the droplets prone to aggregation [42]. The higher values of EAI and ESI in TMP and GPMP may depend on their higher solubility causing more proteins to be adsorbed at the oil–water interface. In addition, the three MPs all exhibited a higher EAI than the mussel MPs due to the differences in species [23].

There was no significant difference in the FA and FS of TMP and GPMP and no significant difference in the FS of three MPs (Figure 3C,D). Results showed that the FA of STMP was significantly higher than that in TMP and GPMP (*p* < 0.05), suggesting that STMP was more capable of forming bubbles. It was reported that the hydrophobic group played a vital role in adsorbing proteins to the air–water interface; the higher surface hydrophobicity made more proteins adsorb to the air–water interface to improve the ability of protein dispersion to trap bubbles in the system leading to forming more stable bubbles [43,44]. In addition, the low content of reactive SH has also been reported to contribute to enhancing the foaming properties of proteins [45]. Therefore, the higher FA of STMP was possibly due to its more hydrophobic sites on the surface and less content of reactive SH.

### 3.8. Comparison of the N-Glycoproteome between MPs

Glycopeptides derived from MPs were enriched with lectin mixtures and deglycosylated by PNGase F in H_2_^18^O. The acetylation of protein N-terminal, oxidation of methionine and deamidation ^18^O of asparagine were set as variable modifications and carbamidomethylation of cysteines was defined as a fixed modification for database searching [46].

The post-translational modification of proteins, especially glycosylation, plays a vital role in the molecular structure, physicochemical properties and functional properties of proteins [47]. As crucial structural components, N-glycans share a common core structure consisting of two N-acetylglusosamines and three mannoses, which have three common types of high-mannose, hybrid and complex. It was reported that the initial glycoform was Glc3Man9GlcNAc2 in the biosynthesis process and other monosaccharides were cleaved or linked to form the final glycoform due to various glycosidases [48]. N-glycans affect the stability of the proteins and other physicochemical properties. N-glycans protect proteins from degradation, oxidation, aggregation, and thermal deformation due to their natural high hydrophilicity [49]. There were 23, 85 and 22 N-glycoproteins identified in TMP, GPMP and STMP (Appendix A). Twenty-four unique glycopeptides, which contained 28 N-glycosylation sites, were identified in TMP. As for GPMP, 113 unique glycopeptides, which had 129 N-glycosylation sites, were identified. Furthermore, in STMP, there were 25 unique glycopeptides that contained 35 N-glycosylation sites (Appendix A). Previous work has reported that all the identified glycopeptides in the three MPs were identified in high precision, and the precursor tolerance was less than 6 ppm (Figure 4). Previous studies have reported that recombinant human IFN-β had high solubility due to containing a homogeneous N-glycosylation site that inhibited disulfide binding and protein aggregation [50]. Compared with STMP, GPMP had more N-glycoproteins and N-glycosylated sites, which was possibly the reason for GPMP’s higher solubility and EAI.

Appendix A shows that for TMP, most of the N-glycoproteins (18) had only one N-glycosylation site, and the other five N-glycoproteins carried two sites. In GPMP, most of the identified N-glycoproteins (57) carried a single N-glycosylation site and other N-glycoproteins (28) contained multiple sites. The most heavily N-glycosylated proteins in GPMP were laminin subunit gamma 1 and an uncharacterized N-glycoprotein (A0A3B4WJQ0), which both contained seven N-glycosylation sites. In STMP, 14 N-glycoproteins contained one N-glycosylation site; eight N-glycoproteins carried multiple sites. The most heavily N-glycosylated proteins in STMP were BCL9 domain-containing proteins and three uncharacterized proteins (A0A3B4A8B5, A0A3B4AKI7 and A0A3B3ZCY4), which all contained three N-glycosylation sites.

There were similarities and differences in identified N-glycoproteins among three MPs. Because the N-glycoproteome analysis of TMP, GPMP and STMP was the first time, some N-glycoproteins identified were uncharacterized. After removing the uncharacterized ones, 18, 77 and 12 unique N-glycoproteins were identified in TMP, GPMP and STMP, respectively (Figure 4D). However, there were two N-glycoproteins, including decorin and carboxylic ester hydrolase, in all three MPs. It was interesting that decorin was usually mentioned and studied in pathology [51,52]. Decorin was a member of the small leucine-rich proteoglycan (SLRP) family. The core protein was covalently attached to the serine residue with the chondroitin/dermatan sulfate glycan chain [53]. It was reported that decorin played an essential role in the fibrillar organization [54]. Although decorin was identified as N-glycoprotein in all three MPs, the locations of the N-glycosylation site of decorin were different. Meanwhile, there were another two common N-glycoproteins, including hemopexin and C-type lectin domain-containing protein in TMP and GPMP. In addition, another N-glycoprotein named integrin beta was identified both in GPMP and STMP.

The types of glycosylated proteins and the number of glycosylated sites in the three MPs varied greatly. The glycosylation modification information was crucial for analyzing different MPs’ structures, functions, and biological activity.

## 4. Conclusions

In this study, MPs’ microstructures, physicochemical properties and functional properties from three different fish breeds were compared, and the N-glycoproteins and N-glycosylation sites of MPs were identified. The SEM observation indicated that all MPs exhibited a similar fibrous structure. The solubility of TMP was the highest among the three MPs (*p* < 0.05). The results of surface hydrophobicity, SH content, intrinsic fluorescence spectrum and FTIR indicated that the protein structures of TMP and GPMP were more folded and stable than STMP. Due to the low reactive SH content and high surface hydrophobicity, STMP exhibited a high FA. The EAI of TMP was the highest (*p* < 0.05), indicating its potential as a stable emulsifier. Twenty-three, 85 and 22 N-glycoproteins that contained 28, 129 and 35 N-glycosylation sites were identified in TMP, GPMP and STMP. GPMP had more N-glycoproteins and N-glycosylated sites, possibly the reason for GPMP’s higher solubility and EAI. It was a first time attempt to analyze MP glycoproteome in different fish species to identify as many N-glycosylation sites as possible. Qualitative N-glycoproteomics analysis of three MPs provides new insights into the analysis of differences in functional properties of fish proteins. The results also provide essential information for better utilization of different fish resources and further understanding of the structures, function and biological activity of different MPs’ glycoproteins.

## Figures and Tables

**Figure 1 foods-11-01705-f001:**
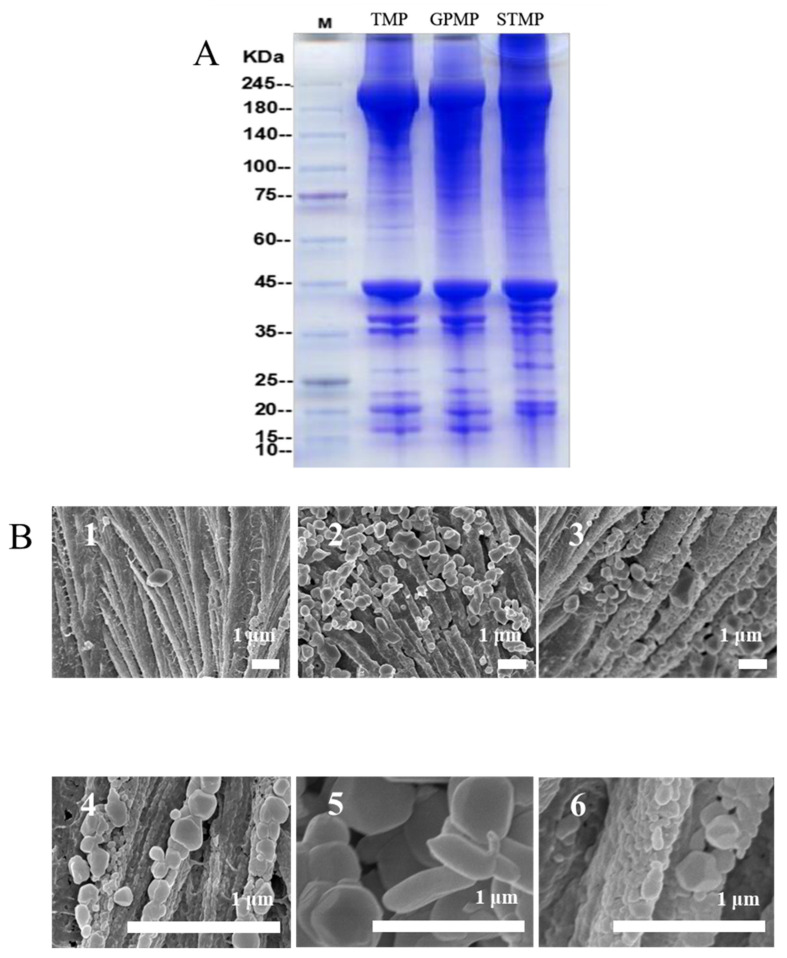
SDS-PAGE electrophoresis (**A**) and the microstructure of MPs observed by scanning electron microscope (SEM) (**B**) (Scale bar = 1 μm) ((**B1**,**B4**) TMP, (**B2**,**B5**) GPMP, (**B3**,**B6**) STMP; (**B1**–**B3**) at the magnification of ×4000, (**B4**–**B6**) at the magnification of ×25,000).

**Figure 2 foods-11-01705-f002:**
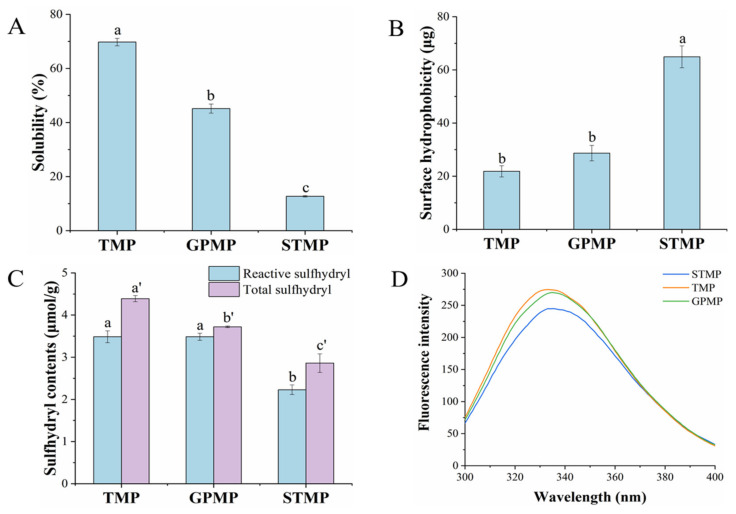
Solubility (**A**), surface hydrophobicity (**B**), total and reactive sulfhydryl (SH) group content (**C**) and intrinsic fluorescence spectra (**D**) of MPs. Different lowercase letters indicate significant differences at *p* < 0.05.

**Figure 3 foods-11-01705-f003:**
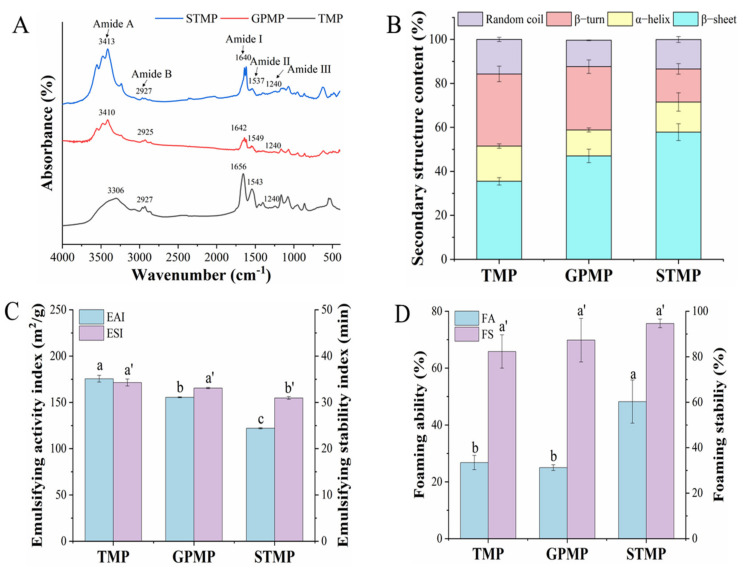
FTIR spectra (**A**), secondary structure contents (**B**), emulsion properties (**C**) and foaming properties (**D**) of MPs. Different lowercase letters indicate significant differences at *p* < 0.05.

**Figure 4 foods-11-01705-f004:**
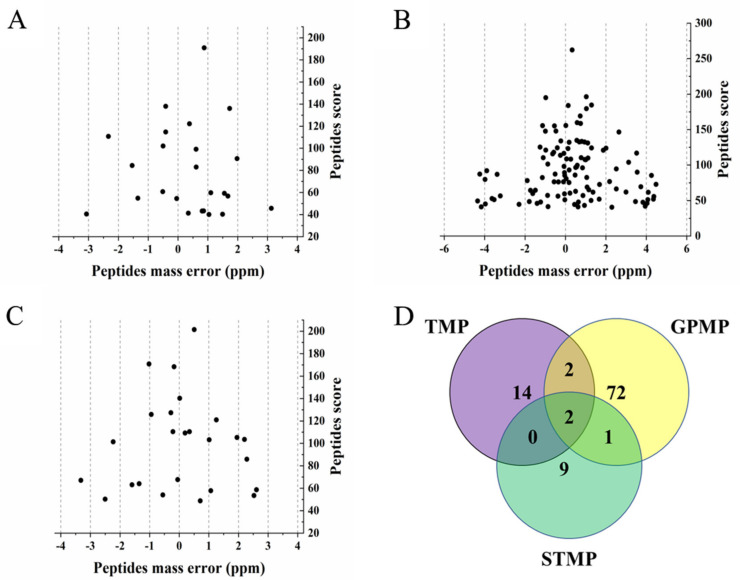
Mass error distribution of identified N-glycopeptides ((**A**) TMP, (**B**) GPMP and (**C**) STMP) and overlapping Venn diagram of identified N-glycoproteins (**D**).

## Data Availability

Not applicable.

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
