# Peer review of "Comparative Study on the Characterization of Myofibrillar Proteins from Tilapia, Golden Pompano and Skipjack Tuna"

_foods, 2022, doi:10.3390/foods11121705_

Round 1

Reviewer 1 Report

Totally speaking, this study is well designed. Firstly, presented results provide useful information for the effective utilization of various fish products, especially fish feed proteins. The physicochemical properties, functional properties and N-glycoproteome of tilapia myofibrillar protein, golden pompano myofibrillar protein and skipjack tuna myofibrillar protein were assessed. The molecular structure of analyzed those three proteins are very well characterized by structural analysis, like as SEM, FTIR, and tertiary structure analysis. The techno-functional properties of the isolated proteins were determined with the proper fluorescence and UV-Vis spectrophotometer assays. This study was suitable for this journal. However, there are few issues needed to be clarified. 

1. Whether the described extraction process of those three fish proteins could be used in the food/feed industry? Please describe the complete extraction procedure in the section of the method so that readers can better understand and have the opportunity to reproduce the same experiments. So, first describe at the preparation of samples for extraction, as from the fillet of fish obtained the starting substrate.

2. What are the yields of extracted / isolated proteins, and what is their concentration?  Explain to the readers what kind of proteins you got, isolates, concentrates or just flour, depending on the protein content. Also, describe the method by which you calculated that concentration, the Kjeldahl or Dumas method?

3. It is advised that the authors recheck the main text during the revision to make this manuscript more readable.

4. The presented results are vary well arganized, also are correctly presented. All experimental results possess the statistical analysis, and major errors in the design itself and the presentation of the results are not noticeable at this time. The authors treated the secondary and tertiary structure of the protein very nicely, all praise. However, there are two issues needed to be clarified.

(1) In the Figure 2, the authors present the basic characteristics of proteins, such as: solubility, SH groups and hydrophobicity, but discuss the results separately. The presented discussion is correct, supported by scientifically available facts, but it would be very interesting for the authors to connect these four functional properties, to find a correlation between them. Thus, between hydrophobicity and solubility and SH groups. In that way, the work would gain even more in the strength and clarity of the results.

(2) Associate the functional properties (EAI, ESI, FA and FS) of the isolated proteins with the previously described properties from the previous point 1, because they are closely related.

Author Response

Reviewer #1:

Totally speaking, this study is well designed. Firstly, presented results provide useful information for the effective utilization of various fish products, especially fish feed proteins. The physicochemical properties, functional properties and N-glycoproteome of tilapia myofibrillar protein, golden pompano myofibrillar protein and skipjack tuna myofibrillar protein were assessed. The molecular structure of analyzed those three proteins are very well characterized by structural analysis, like as SEM, FTIR, and tertiary structure analysis. The techno-functional properties of the isolated proteins were determined with the proper fluorescence and UV-Vis spectrophotometer assays. This study was suitable for this journal. However, there are few issues needed to be clarified.

Response to Reviewer #1:

We truly appreciate your effort in evaluating our manuscript and providing constructive comments and suggestions. We have revised our manuscript accordingly, and the details are as follows. All changes that we have made in the revised manuscript are highlighted in red.

  1. Whether the described extraction process of those three fish proteins could be used in the food/feed industry? Please describe the complete extraction procedure in the section of the method so that readers can better understand and have the opportunity to reproduce the same experiments. So, first describe at the preparation of samples for extraction, as from the fillet of fish obtained the starting substrate.

Response to comment 1:

Thank you. The extraction process of myofibrillar protein (MP) has the potential to the application in food industry. This method of MP extraction is relatively simple and effective, and the reagents used in the extraction process are common and cheap. We have added more details to the extraction of MPs. Please read lines 77-95.

The MP was extracted as the previous study with minor modifications (Bakry, A.; Huang, J.; Zhai, Y.; Huang, Q. Myofibrillar protein with κ-or λ-carrageenans as novel shell materials for microencapsulation of tuna oil through complex coacervation. Food hydrocolloids 2019, 96, 43-53.). Briefly, the red muscle was removed from the fillets and the remaining white muscle was ground using a QSJ-B02R1 commercial grinder (Bear Co., Ltd., China) in ice water. The minced white muscle was rinsed three times in a low phosphate buffer solution (0.05 mol/L NaCl, 3.38 mmol/L NaH2PO4•2H2O, 15.5 mmol/L Na2HPO4•12H2O, pH 7.5) at 4 ℃. Subsequently, the precipitates of tilapia and golden pompano were centrifuged at 7000 g using a refrigerated centrifuge (X1R, Thermo, Osterode, Germany) in high phosphate buffer solution (0.6 mol/L NaCl, 3.38 mmol/L NaH2PO4•2H2O, 15.5 mmol/L Na2HPO4•12H2O, pH 7.0); the precipitate of skipjack tuna was centrifuged at the same condition in high phosphate buffer solution (1.5 mol/L NaCl, 3.38 mmol/L NaH2PO4•2H2O, 15.5 mmol/L Na2HPO4•12H2O, pH 7.0) and then stored at 4 ℃ for 24 h. The mixture was centrifuged for 10 min at 16000 g, the supernatant was collected in the cold distilled water (5-fold) and precipitated at 4 ℃ for 30 min. The precipitate (MP) was collected after two more centrifugations at 16000 g for 15 min. The yield of the extracted MP was 26.90 ± 1.37% and the concentrations of tilapia myofibrillar protein (TMP), golden pompano myofibrillar protein (GPMP) and skipjack tuna myofibrillar protein (STMP) was 11.38 ± 0.69%, 18.78 ± 0.93% and 10.23 ± 0.52%, respectively. The extracted MPs were isolated according to the protein contents. The protein concentration was determined according to the method described by the previous study (Lowry, O.; Rosebrough, N.; Farr, A.L.; Randall, R. Protein measurement with the Folin phenol reagent. Journal of biological chemistry 1951, 193, 265-275.), using bovine serum albumin as a standard.

  1. What are the yields of extracted / isolated proteins, and what is their concentration? Explain to the readers what kind of proteins you got, isolates, concentrates or just flour, depending on the protein content. Also, describe the method by which you calculated that concentration, the Kjeldahl or Dumas method?

Response to comment 2:

Thank you. The yield of the extracted MP was 26.90 ± 1.37% and the concentrations of tilapia myofibrillar protein (TMP), golden pompano myofibrillar protein (GPMP) and skipjack tuna myofibrillar protein (STMP) was 11.38 ± 0.69%, 18.78 ± 0.93% and 10.23 ± 0.52%, respectively. The extracted MPs were isolated according to the protein contents. And the protein concentration was determined according to the method described by Lowry (Lowry, O.; Rosebrough, N.; Farr, A.L.; Randall, R. Protein measurement with the Folin phenol reagent. Journal of biological chemistry 1951, 193, 265-275.), using bovine serum albumin as a standard.

  1. It is advised that the authors recheck the main text during the revision to make this manuscript more readable.

Response to comment 3:

Thank you. We have rechecked to make sure of the readability of the manuscript.

  1. The presented results are very well organized, also are correctly presented. All experimental results possess the statistical analysis, and major errors in the design itself and the presentation of the results are not noticeable at this time. The authors treated the secondary and tertiary structure of the protein very nicely, all praise. However, there are two issues needed to be clarified.

(1) In the Figure 2, the authors present the basic characteristics of proteins, such as: solubility, SH groups and hydrophobicity, but discuss the results separately. The presented discussion is correct, supported by scientifically available facts, but it would be very interesting for the authors to connect these four functional properties, to find a correlation between them. Thus, between hydrophobicity and solubility and SH groups. In that way, the work would gain even more in the strength and clarity of the results.

Response to comment 4 (1):

Thank you. We have added some sentences to connect the four physicochemical properties in the revised manuscript. Please read lines 246-249, 260-261, 266-267, 269-271, and 275-277.

As for myosin, the surface hydrophobicity was inversely proportional to its solubility (You, J.; Pan, J.; Shen, H.; Luo, Y. Changes in physicochemical properties of bighead carp (Aristichthys mobilis) actomyosin by thermal treatment. International Journal of Food Properties 2012, 15, 1276-1285; Li, S.; Zheng, Y.; Xu, P.; Zhu, X.; Zhou, C. L-Lysine and L-arginine inhibit myosin aggregation and interact with acidic amino acid residues of myosin: The role in increasing myosin solubility. Food Chemistry 2018, 242, 22-28.). According to Figure 2A and Figure 2B, the strong hydrophobic force made STMP particles aggregate more compact, thus greatly lowering their solubility.

The unfolding of STMP exposed more SH groups and more internal hydrophobic groups resulting in its higher surface hydrophobicity.

Therefore, more myosin aggregated held by disulfide bonds in STMP also led to the protein aggregation at the top of the electrophoretic pattern and its lowest solubility.

The intrinsic fluorescence intensity is related to hydrophobic residues such as tryptophan and tyrosine, and the intrinsic fluorescence spectrum reflects the changes in the tertiary structure of the protein (Peng, Z.; Zhu, M.; Zhang, J.; Zhao, S.; He, H.; Kang, Z.; Ma, H.; Xu, B. Physicochemical and structural changes in myofibrillar proteins from porcine longissimus dorsi subjected to microwave combined with air convection thawing treatment. Food Chemistry 2021, 343, 128412.).

Therefore, low fluorescence intensity indicated that more hydrophobic residues were exposed to the surface and quenched, leading to the higher surface hydrophobicity of STMP.

(2) Associate the functional properties (EAI, ESI, FA and FS) of the isolated proteins with the previously described properties from the previous point 1, because they are closely related.

Response to comment 4 (2):

Thank you. We have added some sentences to connect the physicochemical properties and functional properties in the revised manuscript. Please read lines 310-317, 321-322 and 331-337.

The exposure of internal hydrophobic groups and sulfhydryl groups due to the unfolding of the protein structure would strengthen the surface hydrophobicity and the formation of disulfide bonds. The enhanced interaction might improve steric stability against the flocculation and aggregation of oil droplets (Ma, W.; Wang, J.; Wu, D.; Xu, X.; Wu, C.; Du, M. Physicochemical properties and oil/water interfacial adsorption behavior of cod proteins as affected by high-pressure homogenization. Food Hydrocolloids 2020, 100, 105429.). However, even though the value of surface hydrophobicity of STMP was the highest and reactive SH content of STMP was the lowest, it was interestingly found that the emulsifying properties of STMP were not improved with more hydrophobic sites.

The higher values of EAI and ESI in TMP and GPMP may depend on their higher solubility making more proteins adsorb at the oil-water interface.

It was reported that the hydrophobic group played a vital role in adsorbing proteins to the air-water interface; the higher surface hydrophobicity made more proteins adsorb to the air-water interface to improve the ability of protein dispersion to trap bubbles in the system leading to form more stable bubbles (Jiang, J.; Wang, Q.; Xiong, Y.L. A pH shift approach to the improvement of interfacial properties of plant seed proteins. Current opinion in food science 2018, 19, 50-56; Alavi, F.; Chen, L.; Wang, Z.; Emam-Djomeh, Z. Consequences of heating under alkaline pH alone or in the presence of maltodextrin on solubility, emulsifying and foaming properties of faba bean protein. Food Hydrocolloids 2021, 112, 106335). In addition, the low content of reactive SH has also been reported to contribute to enhancing the foaming properties of proteins (Li, P.; Sun, Z.; Ma, M.; Jin, Y.; Sheng, L. Effect of microwave-assisted phosphorylation modification on the structural and foaming properties of egg white powder. LWT 2018, 97, 151-156.). Therefore, the higher FA of STMP was possibly due to its more hydrophobic sites on the surface and less content of reactive SH.

Reviewer 2 Report

The authors reported the comparative study of three different sources of myofibrillar protein (tilapia, golden pompano, and skipjack) based on their N-glycoproteome, physicochemical and functional properties.

The experiment is well designed and the results and conclusions were contrasted with the literature. However, some issues need to be addressed before publication:

-          It would be important to include a blank or control in order to compare the results and techniques against a known sample.

Make the conclusions shorter and avoid the discussion of results. They were already discussed in results and discussions section. It will be good idea to focus in the scope of the experiment, how these results addressed the main objective, possible applications, future works, etc.

Author Response

Reviewer #2:

The authors reported the comparative study of three different sources of myofibrillar protein (tilapia, golden pompano, and skipjack) based on their N-glycoproteome, physicochemical and functional properties. The experiment is well designed and the results and conclusions were contrasted with the literature. However, some issues need to be addressed before publication:

Response to Reviewer #2:

We truly appreciate your effort in evaluating our manuscript and providing constructive comments and suggestions. We have revised our manuscript accordingly, and the details are as follows. All changes that we have made in the revised manuscript are highlighted in red.

  1. It would be important to include a blank or control in order to compare the results and techniques against a known sample.

Response to comment 1:

Thank you. We have added some previous studies to compare with our results. Please read lines 226-227, 242-244, 255-257 and 322-323.

And the solubility of TMP was close to the cod proteins in the previous study (Ma, W.; Wang, J.; Wu, D.; Xu, X.; Wu, C.; Du, M. Physicochemical properties and oil/water interfacial adsorption behavior of cod proteins as affected by high-pressure homogenization. Food Hydrocolloids 2020, 100, 105429.).

A previous study reported similar results that the surface hydrophobicity of the tuna sample was the highest among all samples (Dara, P.K.; Geetha, A.; Mohanty, U.; Raghavankutty, M.; Mathew, S.; Nagarajarao, R.C.; Rangasamy, A. Extraction and characterization of myofibrillar proteins from different meat sources: a comparative study. Journal of Bioresources and Bioproducts 2021, 6, 367-378.).

It was found that the SH group contents of all 3 MPs were lower than the results from grass carp in the previous study, which was probably due to the differences in species (Xiong, G.; Cheng, W.; Ye, L.; Du, X.; Zhou, M.; Lin, R.; Geng, S.; Chen, M.; Corke, H.; Cai, Y.-Z. Effects of konjac glucomannan on physicochemical properties of myofibrillar protein and surimi gels from grass carp (Ctenopharyngodon idella). Food Chemistry 2009, 116, 413-418.).

In addition, the 3 MPs all exhibited a higher EAI than the mussel MPs due to the differences in species (Cha, Y.; Shi, X.; Wu, F.; Zou, H.; Chang, C.; Guo, Y.; Yuan, M.; Yu, C. Improving the stability of oil-in-water emulsions by using mussel myofibrillar proteins and lecithin as emulsifiers and high-pressure homogenization. Journal of Food Engineering 2019, 258, 1-8.).

2.Make the conclusions shorter and avoid the discussion of results. They were already discussed in results and discussions section. It will be good idea to focus in the scope of the experiment, how these results addressed the main objective, possible applications, future works, etc.

Response to comment 2:

Thank you. We have rewritten the conclusions according to your suggestion.

Conclusions

In this study, MPs' microstructures, physicochemical properties, and functional properties from 3 different fish breeds were compared, and the N-glycoproteins and N-glycosylation sites of MPs were identified. The SEM observation indicated that all MPs exhibited a similar fibrous structure. The solubility of TMP was the highest among 3 MPs (p < 0.05). The results of surface hydrophobicity, SH content, intrinsic fluorescence spectrum and FTIR indicated that the protein structures of TMP and GPMP were more folded and stable than STMP. Due to the low reactive SH content and high surface hydrophobicity, STMP exhibited a high FA. The EAI of TMP was the highest (p < 0.05), indicating its potential as a stable emulsifier. 23, 85 and 22 N-glycoproteins that contained 28, 129 and 35 N-glycosylation sites were identified in TMP, GPMP and STMP. GPMP had more N-glycoproteins and N-glycosylated sites, possibly the reason for GPMP's higher solubility and EAI. It was the first time attempt to analyze MP glycoproteome in different fish species to identify as many N-glycosylation sites as possible. Qualitative N-glycoproteomics analysis of 3 MPs provides new insights into the analysis of differences in functional properties of fish proteins. The results also provide essential information for better utilization of different fish resources and further understanding of the structures, function, and biological activity of different MPs' glycoproteins.